# Thrombotic Events after COVID-19 Vaccination in the Over-50s: Results from a Population-Based Study in Italy

**DOI:** 10.3390/vaccines9111307

**Published:** 2021-11-10

**Authors:** Ileana Baldi, Danila Azzolina, Andrea Francavilla, Patrizia Bartolotta, Giulia Lorenzoni, Diego Vanuzzo, Dario Gregori

**Affiliations:** 1Unit of Biostatistics, Epidemiology and Public Health, Department of Cardiac Thoracic Vascular Sciences and Public Health, University of Padova, 35121 Padova, Italy; andrea.francavilla@studenti.unipd.it (A.F.); patrizia.bartolotta@studenti.unipd.it (P.B.); giulia.lorenzoni@unipd.it (G.L.); dario.gregori@unipd.it (D.G.); 2Department of Medical Science, University of Ferrara, 44100 Ferrara, Italy; danila.azzolina@unife.it; 3Cardiovascular Prevention Centre, 33100 Udine, Italy; diego50rdc@gmail.com

**Keywords:** claims data, COVID-19, Vaxzevria, safety, adverse events, Italy

## Abstract

Several European countries suspended or changed recommendations for the use of Vaxzevria (AstraZeneca) for suspected adverse effects due to atypical blood-clotting. This research aims to identify a reference point towards the number of thrombotic events expected in the Italian population over 50 years of age who received Vaxzevria from 22 January to 12 April 2021. The venous thromboembolism (VT) and immune thrombocytopenia (ITP) event rates were estimated from a population-based cohort. The overall VT rate was 1.15 (95% CI 0.93–1.42) per 1000 person-years, and the ITP rate was 2.7 (95% CI 0.7–11) per 100,000 person-years. These figures translate into 83 and two expected events of VT and ITP, respectively, in the 15 days following the first administration of Vaxzevria. The number of thrombotic events reported from the Italian Medicines Agency does not appear to have increased beyond that expected in individuals over 50 years of age.

## 1. Introduction

Starting in March 2021, several European countries suspended or changed recommendations for the use of Vaxzevria (formerly vaccine AstraZeneca), an adenoviral vector vaccine for preventing coronavirus disease 2019 (COVID-19) due to an increasing number of unusual blood clots.

After an in-depth analysis, the European Medicines Agency (EMA) drug safety committee concluded that the vaccine was safe and was not generally associated with a higher risk of blood-clotting. However, it could not rule out a link with two very rare and severe clotting conditions and stated these rare side effects should be listed in the vaccine’s side effects [1].

Shultz and colleagues [2] compared the clinical picture of cases presented with severe thrombosis and thrombocytopenia after receiving the first dose of Vaxzevria to that of heparin-induced thrombocytopenia. They proposed that these cases represent a vaccine-related variant of spontaneous heparin-induced thrombocytopenia, referred to as vaccine-induced immune thrombotic thrombocytopenia (VITT).

No major safety concerns were identified in the initial trial [3], which involved nearly 24,000 adults. Even side effects that are uncommon during clinical trials may affect a relatively large number of people once vaccines are widely distributed. Thus, it is not surprising that new reports of side effects have emerged from the real-world data after millions of Vaxzevria doses have been administered.

In Italy, the first doses of Vaxzevria arrived in February 2021, and the Italian Medicines Agency (AIFA) approved administration to individuals 55 years of age and older. Following concerns of a link to blood clots in younger people, Italy now reserves Vaxzevria for the over-60s.

Nevertheless, elderly patients are poorly represented in COVID-19 clinical trials and vaccine trials [4]. Hence, there is limited evidence and knowledge about the efficacy and safety of drugs and vaccines in this population, particularly vulnerable to COVID-19 [5].

This study aims to quantify the order of magnitude of the number of blood clot events, from ordinary to very rare types, expected in the Italian population over 50 years of age who received Vaxzevria. To this purpose, age and gender-specific venous thromboembolism (VT) event rates and overall immune thrombocytopenia (ITP) event rate were estimated from a population-based cohort with a 37-year follow-up and applied to the vaccinated with the first dose of Vaxzevria.

## 2. Materials and Methods

### 2.1. Data Sources

The “Martignacco project” (MP), promoted by the World Health Organization (WHO), was started in 1977 to evaluate the impact of health promotion interventions on cardiovascular health in middle-aged (40–59 years) subjects within a geographically defined community in the northeast of Italy. The population-based cohort data for the present analysis referred to the 3066 subjects enrolled in 1977 and followed up through 31 December 2014, through a computerized record linkage system with administrative sources on healthcare use (details given elsewhere [6,7,8]). All Italian citizens have equal access to health care. Thus, the inequality in health care access is not a matter of insurance status.

The numbers of Vaxzevria administrations stratified by age and gender were retrieved from the official and freely accessible Italian repository of the COVID-19 vaccine data [9]. The administration period from 22 January to 12 April 2021, was chosen to capture only the first dose per subject. Thus, in this period, the number of doses coincides with the number of vaccinated subjects.

The number of side effects was then retrieved from the reports of suspected adverse reactions to COVID-19 vaccines submitted to the Italian Pharmacovigilance Network [10]. The fourth surveillance report, covering 22 January to 26 April 2021 was used. A 2 week-difference between the end of Vaxzevria administration period and the release date of the report was intentionally assumed to ensure a minimum safety follow-up.

### 2.2. VT and ITP Events Identification

A VT event in the MP cohort is defined as the first occurrence in the study period of hospitalization with a primary or secondary diagnosis of pulmonary embolism (415.1) or deep vein thrombosis (451.11, 451.19, 451.2, 451.81, and 453.8), according to the International Classification of Diseases, 9th Revision, Clinical Modification (ICD-9-CM). As reported in a literature review [11], this validated algorithm ensures good predictive values. An ITP event in the MP cohort is defined as the first occurrence in the study period of hospitalization with a primary or secondary diagnosis of immune thrombocytopenic purpura (ICD-9-CM 287.31) or primary thrombocytopenia (ICD-9-CM 287.3) [12]. The latter diagnostic code was used only for hospitalizations that occurred before the ICD-9-CM update 2007 version.

### 2.3. Calculation of Rates and Projections

The VT rates (per 1000 person-years) in the MP cohort were calculated by age group and gender, with 95% confidence intervals (95% CI). For this purpose, age at hospitalization was split into 10-year increment age groups, from 50 to 80+ years. The rates’ denominators were computed in years of observation beginning in June 1977 until a VT event or the end of follow-up.

The ITP number of events and rates was also computed in the MP cohort. Due to the rarity of the disease, the event rate was reported overall without stratifying by demographic categories.

Age and gender-specific VT rates were applied to the Italian population vaccinated with Vaxzevria from 22 January to 12 April 2021 to estimate the number of expected cases in the 15 days following vaccination. The same operation was carried out to estimate the ITP events expected in the whole vaccinated population.

Expected side effects were computed by multiplying the number of vaccinated people for each demographic category (age, gender) by the VT and ITP rates (calculated in the MP cohort) and by 0.041 (the result of 15 days/365 days) and divided by 1000. The prediction intervals for the expected number of events were computed via a 10,000 bootstrap resampling procedure. The expected number of events were iteratively resampled from a Poisson random variable whose parameters were given by the expected event rate calculated for the Martignacco reference cohort. The 95% prediction interval bounds were derived by considering the 0.025 and 0.975 quantiles of the resampled distribution.

## 3. Results

### 3.1. VT and ITP Rates in the Martignacco Project Cohort

The overall VT rate in the MP cohort was 1.15 (95% CI 0.93–1.42) events per 1000 person-years. The VT rates increased with age for both genders. A lower VT rate was observed in females under 69 years in comparison with males. The situation was reversed in the over-70s (Table 1).

Two ITP events were observed in the MP cohort over 72,293 person-years, resulting in an ITP rate of 2.7 (95% CI 0.7–11) cases over 100,000 person-years.

### 3.2. Expected VT and ITP Events in the Vaccinated with Vaxzevria

From 22 January to 12 April, 2,702,730 doses of the Vaxzevria were administered. Among them, 65.4% (1,767,281 doses) were delivered in those over 50 years old. During this time interval, based on the extracted VT and ITP rates from the Martignacco data (Table 1), the overall expected cases of VT and ITP were 83 (Table 1) and 1.8 (95% CI 0.46–6.68), respectively.

## 4. Discussion

Blood clots are more common in older people and might not trigger the same attention and in-depth investigation that blood clots in younger Vaxzevria recipients have had. This study quantifies the order of magnitude of a very rare blood clot type, ITP, and a common one, VT, expected in the Italian population aged over 50 years of age following the administration of Vaxzevria.

To our knowledge, this is one of the few studies that has assessed VT and ITP rates within a population-based cohort and not through hospital episode-based statistics. According to rate projections calculated in those over-50s in the Martignacco cohort, on a national basis a total of 83 VT (54% in women) and two ITP events is expected in the 15 days following vaccination in those over 50 years old.

As to VT estimates, other studies relying on episode-based statistics, reported similar findings based on pre-pandemic incidence rates from the entire Danish population [13]. Regarding ITP, the estimated rate of 2.7 per 100,000 person-years is in line with the results of population-based studies in France, UK, and Japan, reporting an ITP incidence ranging from 2.2 to 3.9 per 100,000 person-years [14].

On 10 May the AIFA has published the fourth COVID-19 Vaccine Surveillance Report [10]. The report includes suspected adverse reactions registered in the National Pharmacovigilance Network between 27 December 2020 and 26 April 2021 for the four vaccines used in the current vaccination campaign. Twenty-nine cases of cerebral venous thrombosis and five cases of venous thrombosis of atypical location following Vaxzevria administration were reported. Of these, 22 cases (65%) affected women with an average age of about 48 years, and 12 (35%) affected men with an average age of about 52 years. The mean time to onset was approximately 8 days after administration of the first dose of Vaxzevria vaccine.

The lack of public disclosure of adverse reaction attributes at the individual level (i.e., age, gender, diagnostic subtypes, severity) captured by the National Pharmacovigilance Network hampers the comparison between these figures (*n* = 34) and the expected events (*n* = 85). However, caution should be observed in making this comparison because reporting of side effects is susceptible to biases and misclassifications. For example, a complex syndrome such as VITT could be misclassified, particularly before European regulators expressed concerns about a possible link between rare blood clots and Vaxzevria. Now that the concern about this potential association has become public, clinicians will be on the lookout for it, and reporting could increase.

Nevertheless, study findings suggest that the number of thrombotic events following Vaxzevria administration does not appear to have increased beyond that expected in those over 50 years of age.

In a press release dated 8 April 2021, the EMA stated that a causal relationship between the vaccination with Vaxzevria and VITT could be a reasonable possibility. The regulatory agency also declared that other studies will be conducted to identify the exact pathophysiological mechanism for these thrombotic events and define the precise extent of the risk [15].

The prompt development of a safe and effective vaccine to reduce the spread of COVID-19 and establish higher levels of herd immunity is a global imperative.

In light of the above results, it must be underlined that thrombotic events were not uncommon in the pre-pandemic era—especially with increasing age—as appears in the Martignacco cohort.

Therefore, it should be expected that these kinds of events will occur in the older population who get vaccinated, representing the result of a physio-pathological phenomenon that naturally takes place rather than the direct consequence of the vaccine administration.

In conclusion, it is safe to say that thrombosis occurs relatively often and that the incident cases following the vaccine should not be considered a priori as adverse effects of the vaccine itself.

Although our findings should be interpreted in the context of the strengths and limitations of the use of claims in outcome research [11], there is a reason to be optimistic about the safety of Vaxzevria in a population usually under-represented in vaccine trials.

These findings contribute to the ongoing discussion on vaccine safety, particularly for vaccines that rely on adenoviruses where occurrence of severe blood clotting and VITT constitute a significant concern.

## Figures and Tables

**Table 1 vaccines-09-01307-t001:** Number of first hospitalizations for venous thromboembolism (VT), person-years (PY), the rates (×1000), and 95% confidence intervals (95% CI) according to gender and age classes. Number of administered doses of Vaxzevria in Italy as of 12 April 2021. Expected VT events in the 15 days following vaccination with 95% prediction intervals (95% PI).

		Martignacco Cohort				COVID-19 Vaccinated Cohort		
Gender	Age	VT	PY	Rate	(95%CI)	Doses	Expected VTs	(95% PI)
**Female**	50–59	2	6405	0.312	(0.078–1.248)	350,064	4.49	(3.69–5.33)
	60–69	10	13,460	0.743	(0.400–1.381)	215,887	6.59	(5.58–7.63)
	70–79	22	12,802	1.718	(1.131–2.610)	404,790	28.58	(26.40–30.63)
	≥80	10	6275	1.594	(0.857–2.962)	36,940	2.42	(1.85–3.03)
**Male**	50–59	5	6422	0.779	(0.324–1.871)	228,011	7.30	(6.23–8.41)
	60–69	13	12,393	1.049	(0.609–1.807)	148,207	6.39	(5.37–7.38)
	70–79	15	10,418	1.440	(0.868–2.388)	358,223	21.20	(19.35–23.00)
	≥80	6	4117	1.458	(0.655–3.244)	25,159	1.51	(1.03–2.01)
**Overall**		83	72,293	1.148	(0.926–1.424)	1,767,281	83.38 *	(79.58–86.88)

* The global expected number of events was computed by considering the overall estimated incidence rate 1.148.

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
