# Peer review of "Thrombotic Events after COVID-19 Vaccination in the Over-50s: Results from a Population-Based Study in Italy"

_vaccines, 2021, doi:10.3390/vaccines9111307_

Round 1
Reviewer 1 Report
Review of Baldi et al. “Thrombotic events after COVID-19 vaccination in the over 50s results from a population-based study in Italy.”
Vaccines 2021
In this manuscript the authors bring to light some healthcare data collected over a long time by the World Health Organization under the “Martignacco Project”(MP). Authors have utilized this rich data source to calculate the rates of Venuous Thromboemolism (VT) and Immune Thrombocytopenia (ITP) in general middle aged Italian population. Authors have compared these rates to the those observed as “adverse effects due to atypical blood clotting” after administering the vaccine Vaxzevria to protect from the SARS-CoV2 infection. Starting in March 2021, this led many European countries to suspend the use of or change recommendations for the use of Vaxzevria.
Authors observe that overall VT and ITP rates calculated from the MP project data are very similar to the rates observed in the 15 days following the first administration of Vaxzevria and that there are no additional thrombotic events attributed to the vaccine administration.
- Authors have contributed to the ongoing discussion 20 on safety for vaccines where immune thrombotic thrombocytopenia is a major concern.
- This study is important evidence showing that majority of the thrombotic events associated with Vaxzevria usage were pre-existing and independent to the vaccine usage.
- Authors have carefully assured that the long term thrombotic events data that they gathered from the MP project was controlled for socioeconomic backgrounds of the participants.
- The findings from this manuscript are important. However, the authors have underutilized their results and not discussed efficiently that the benefits of vaccines are significantly disregarded by incorrectly attributing the thrombotic events to vaccine usage. Authors should discuss this more explicitly.
Author Response
We thank the reviewer for pointing out this issue. We modified the text accordingly.
Reviewer 2 Report
This paper estimates the background rate of thrombotic events in the Italian population. This rate is important because the AstraZeneca vaccine has been proposed to cause such events. If the rate after vaccination is no higher than the background rate, it would imply that the AstraZeneca vaccine does not cause these events. Which is what the authors observe.
MAJOR
It appears that the overall confidence interval was obtained by summing the bounds of the component confidence intervals. e.g.
1.12
3.55
18.81
1.3
3.04
3.71
12.78
0.68 +
=
44.99
That is not correct. Except in very unusual circumstances, which if is the case here, you need to justify clearly in your methods.
Try Googling "summing confidence intervals" to learn how to do this properly. Or consult a statistician or statistics textbook.
2. "No major safety concerns were identified in the initial trial [3], which involved tens of thousands of adults."
Can you provide an exact number of participants in the initial trial, at least to the nearest ten thousand?
3. "However, it is not surprising that new reports of side effects have emerged from the real-world data after millions of Vaxzevria doses have been administered."
This may not be obvious to some readers. And if it is obvious to all readers there is no need to write it. So you need to write why it is not surprising.
4. Clarify where your uncertainty is coming from.
e.g.,
"The number of VT events and the VT rates (per 1000 person-years) in the MP cohort were calculated by age group and gender, with 95% confidence intervals (95% CI)."
Why is there uncertainty? Aren't you counting events in medical records? That's an absolute number, right? With no uncertainty.
MINOR
ABSTRACT
put the word 'AstraZeneca' somewhere in the Abstract so that automated searches can use that keyword to find this article.
watch for parallel structure throughout the manuscript. e.g.
"per 1000 person-years, and the ITP rate is 2.7 (95% CI 0.7-11) over 100,000 person-years"
should be
"per 1000 person-years, and the ITP rate is 2.7 (95% CI 0.7-11) per 100,000 person-years"
OR
"over 1000 person-years, and the ITP rate is 2.7 (95% CI 0.7-11) over 100,000 person-years"
and then decide what to do with this sentence to keep parallel structure:
"These figures translate into 78 (54% in females) and 2 expected events of VT and ITP"
probably best just to make it
"These figures translate into 78 and 2 expected events of VT and ITP"
rather than something like
"These figures translate into 78 (54% in females) and 2 (~50% in females) expected events of VT and ITP"
and then better yet, just write:
"These figures translate into 78 expected events of VT and 2 expected events of ITP"
be more clear about what "over 50s" means. e.g.
"The number of thrombotic events reported from the Italian Medicines Agency does not appear to have increased beyond expected in individuals over 50 years of age."
Delete this sentence:
"This study contributes to the ongoing discussion on safety for vaccines where immune thrombotic thrombocytopenia is a major concern."
It doesn't really add anything that hasn't been said earlier in the Abstract.
INTRO
"In Italy, the first doses of Vaxzevria arrived in February 2021, and the Italian Medicines Agency (AIFA) extended the administration until 55 years of age."
Since you have talked about indications before, you can't write "extended" now. Maybe just write:
"In Italy, the first doses of Vaxzevria arrived in February 2021, and the Italian Medicines Agency (AIFA) approved administration to individuals 55 years of age and older."
METHODS
VT and ITP events identification
Please clarify the beginning and end dates for ascertainment of VT and ITP events. In "Data Sources" the reader is led to believe that this period will be 1977 to December 31, 2014. However in this section it appears that VT are ascertained from 1977 to whenever the ICD was updated, presumably sometime around 2007. Are the ascertainment end dates the same for both VT and ITP events?
RESULTS
"A lower VT rate is observed in females over 69 years in comparison with males. The situation is reversed in the over 70s"
Hunh?
So 70-year old females have lower rates than males, but 71-year old men have lower rates than females??? I think there is a typo here, and you meant to write "under".
TABLE 1
Is the number of doses = number of people vaccinated? If so, make it clear that is the case. If not, add a column for the number of people vaccinated.
Add a column for the number of people in the Martignacco Cohort for each demographic row. Presumably this is going to be very close the "person years" divided by the number of years of followup, adjusted for cohort attrition.
DISCUSSION
"The mean time to onset was approximately 8 days after administration of the first dose of Vaxzevria vaccine."
How long were the patients followed? 8 days is expected if they were followed for 15 days and the vents were random. If they were followed for a different period of time, maybe the difference from expected randomness implies causality. But could also be related to ascertainment biases.
So if I understand correctly, you are predicting 80 events in the 14 days following vaccination and 34 events were observed, right? Maybe you could write this, immediately before you write comparisons are awkward and hard to interpret.
GENERAL
try pasting the entire text of the document into Google Docs or a grammar checker. It should pick up errors like the double punctuation in this sentence:
"All Italian citizens have equal access to health care:. thus, the inequality in health care access is not a matter of insurance status."
"At the time of writing,"
mention the year/month, at least, because the reader doesn't know.
and you can probably delete this sentence because I am pretty sure two weeks have passed, and more.
"A 2 week-difference between the end of Vaxzevria administration period and the release date of the report has been intentionally assumed to ensure a minimum safety follow-up."
consider using this notation for confidence intervals
345 (339 - 351)
or perhaps, if appropriate
345 (+/- 6)
rather than placing all three values in separate columns of a table.
That way
1. we know what number the confidence interval is for (in this case, 345), which isn't always obvious in your table (e.g., is 0.078 for VT, PY, or Rate)
2. it avoids making it seem like confidence intervals are naturally summable (which could lead to a reader error, even if the author error is corrected).
Author Response
Reviewer 2
This paper estimates the background rate of thrombotic events in the Italian population. This rate is important because the AstraZeneca vaccine has been proposed to cause such events. If the rate after vaccination is no higher than the background rate, it would imply that the AstraZeneca vaccine does not cause these events. Which is what the authors observe.
MAJOR
It appears that the overall confidence interval was obtained by summing the bounds of the component confidence intervals. e.g.
1.12
3.55
18.81
1.3
3.04
3.71
12.78
0.68 +
=
44.99
That is not correct. Except in very unusual circumstances, which if is the case here, you need to justify clearly in your methods.
Try Googling "summing confidence intervals" to learn how to do this properly. Or consult a statistician or statistics textbook.
We completely agree with the reviewer; the confidence interval bounds have been recomputed and reported in Table 1. Specifications concerning the calculation procedure were included in the method section of the manuscript.
- "No major safety concerns were identified in the initial trial [3], which involved tens of thousands of adults."
Can you provide an exact number of participants in the initial trial, at least to the nearest ten thousand?
We revised the text as suggested.
- "However, it is not surprising that new reports of side effects have emerged from the real-world data after millions of Vaxzevria doses have been administered."
This may not be obvious to some readers. And if it is obvious to all readers there is no need to write it. So you need to write why it is not surprising.
We thank the reviewer for this comment. We revised the text accordingly.
- Clarify where your uncertainty is coming from.
e.g.,
"The number of VT events and the VT rates (per 1000 person-years) in the MP cohort were calculated by age group and gender, with 95% confidence intervals (95% CI)."
Why is there uncertainty? Aren't you counting events in medical records? That's an absolute number, right? With no uncertainty.
Thank you for pointing this out. The reviewer is correct.
MINOR
ABSTRACT
put the word 'AstraZeneca' somewhere in the Abstract so that automated searches can use that keyword to find this article.
We thank the reviewer for this suggestion. We revised the text as suggested.
watch for parallel structure throughout the manuscript. e.g.
"per 1000 person-years, and the ITP rate is 2.7 (95% CI 0.7-11) over 100,000 person-years"
should be
"per 1000 person-years, and the ITP rate is 2.7 (95% CI 0.7-11) per 100,000 person-years"
OR
"over 1000 person-years, and the ITP rate is 2.7 (95% CI 0.7-11) over 100,000 person-years"
and then decide what to do with this sentence to keep parallel structure:
"These figures translate into 78 (54% in females) and 2 expected events of VT and ITP"
probably best just to make it
"These figures translate into 78 and 2 expected events of VT and ITP"
rather than something like
"These figures translate into 78 (54% in females) and 2 (~50% in females) expected events of VT and ITP"
and then better yet, just write:
"These figures translate into 78 expected events of VT and 2 expected events of ITP"
We thank the reviewer for these suggestions. We revised the text as suggested.
be more clear about what "over 50s" means. e.g.
"The number of thrombotic events reported from the Italian Medicines Agency does not appear to have increased beyond expected in individuals over 50 years of age."
We thank the reviewer for this suggestion. We revised the text as suggested.
Delete this sentence:
"This study contributes to the ongoing discussion on safety for vaccines where immune thrombotic thrombocytopenia is a major concern."
It doesn't really add anything that hasn't been said earlier in the Abstract.
We thank the reviewer for this suggestion. We revised the text as suggested.
INTRO
"In Italy, the first doses of Vaxzevria arrived in February 2021, and the Italian Medicines Agency (AIFA) extended the administration until 55 years of age."
Since you have talked about indications before, you can't write "extended" now. Maybe just write:
"In Italy, the first doses of Vaxzevria arrived in February 2021, and the Italian Medicines Agency (AIFA) approved administration to individuals 55 years of age and older."
We thank the reviewer for this suggestion. We revised the text as suggested.
METHODS
VT and ITP events identification
Please clarify the beginning and end dates for ascertainment of VT and ITP events. In "Data Sources" the reader is led to believe that this period will be 1977 to December 31, 2014. However in this section it appears that VT are ascertained from 1977 to whenever the ICD was updated, presumably sometime around 2007. Are the ascertainment end dates the same for both VT and ITP events?
We thank the reviewer for pointing out this issue. The ascertainment period actually covered 37 years. Only for hospitalizations that occurred before 2007, we searched for the code ICD-9-CM 287.3.
RESULTS
"A lower VT rate is observed in females over 69 years in comparison with males. The situation is reversed in the over 70s"
Hunh?
So 70-year old females have lower rates than males, but 71-year old men have lower rates than females??? I think there is a typo here, and you meant to write "under".
Thank you for pointing this out. The reviewer is correct.
TABLE 1
Is the number of doses = number of people vaccinated? If so, make it clear that is the case. If not, add a column for the number of people vaccinated.
We intentionally considered a short time frame (< 3 months) from the beginning of Vaxzevria administration to capture only first doses. So there is a 1:1 ratio between doses and individuals.
Add a column for the number of people in the Martignacco Cohort for each demographic row. Presumably this is going to be very close the "person years" divided by the number of years of followup, adjusted for cohort attrition.
In Table1, age refers to age at first hospitalization for VT. Therefore we cannot report the number of people in the Martignacco Cohort by age/gender unless we introduce an age-period-cohort/gender stratification.
DISCUSSION
"The mean time to onset was approximately 8 days after administration of the first dose of Vaxzevria vaccine."
How long were the patients followed? 8 days is expected if they were followed for 15 days and the vents were random. If they were followed for a different period of time, maybe the difference from expected randomness implies causality. But could also be related to ascertainment biases.
This information about mean time to onset was directly taken from the Fourth AIFA Report and based on reports included in the National Pharmacovigilance Network from Dec 2020 to Apr 2021. The analysis at national level was conducted by the “Working Group for the evaluation of thrombotic risks from Covid-19 vaccines”.
So if I understand correctly, you are predicting 80 events in the 14 days following vaccination and 34 events were observed, right? Maybe you could write this, immediately before you write comparisons are awkward and hard to interpret.
The reviewer is correct. We revised the text as suggested.
GENERAL
try pasting the entire text of the document into Google Docs or a grammar checker. It should pick up errors like the double punctuation in this sentence:
"All Italian citizens have equal access to health care:. thus, the inequality in health care access is not a matter of insurance status."
Done.
"At the time of writing,"
mention the year/month, at least, because the reader doesn't know.
and you can probably delete this sentence because I am pretty sure two weeks have passed, and more.
We thank the reviewer for this suggestion. We rephrased the sentence.
"A 2 week-difference between the end of Vaxzevria administration period and the release date of the report has been intentionally assumed to ensure a minimum safety follow-up."
This sentence refers to the fact that we considered adults over 50 years of age vaccinated with Vaxzevria by April 12 to ensure a minimum follow up of 14 days. In fact, the AIFA report relies on reports included in the Pharmacovigilance Network till April 26, 2021. We rephrased the sentence.
consider using this notation for confidence intervals
345 (339 - 351)
or perhaps, if appropriate
345 (+/- 6)
rather than placing all three values in separate columns of a table.
That way
1. we know what number the confidence interval is for (in this case, 345), which isn't always obvious in your table (e.g., is 0.078 for VT, PY, or Rate)
2. it avoids making it seem like confidence intervals are naturally summable (which could lead to a reader error, even if the author error is corrected).
We thank the reviewer for this suggestion. We revised the table accordingly.
Round 2
Reviewer 2 Report
The authors have adequately responded to my comments.